# Global Research on Carbon Emissions: A Scientometric Review

**Lebunu Hewage Udara Willhelm Abeydeera [1],\* 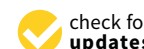, Jayantha Wadu Mesthrige [2] and Tharushi Imalka Samarasinghalage [1]**

[1] Department of Building and Real Estate, The Hong Kong Polytechnic University, Hung Hom 999077, Hong Kong

[2] School of Property, Construction and Project Management, RMIT University, Melbourne 3001, Australia

\* Correspondence: udaraw.lebunuhewage@connect.polyu.hk

**Abstract:** Greenhouse gases such as sulfur dioxide, nitrogen dioxide, and carbon dioxide have been recognized as the prime cause of global climate change, which has received significant global attention. Among these gases, carbon dioxide is considered as the prominent gas which motivated researchers to explore carbon reduction and mitigation strategies. Research work on this domain expands from carbon emission reporting to identifying and implementing carbon mitigation and reduction strategies. A comprehensive study to map global research on carbon emissions is, however, not available. Therefore, based on a scientometric analysis method, this study reviewed the global literature on carbon emissions. A total of 2945 bibliographic records, from 1981 to 2019, were extracted from the Web of Science core collection database and analyzed using techniques such as co-author and co-citation analysis. Findings revealed an increasing trend of publications in the carbon emission research domain, which has been more visible in the past few years, especially during 2016–2018. The most significant contribution to the domain was reported from China, the United States, and England. While most prolific authors and institutions of the domain were from China, authors and institutions from the United States reported the best connection links. It was revealed that evaluating greenhouse gas emissions and estimating the carbon footprint was popular among the researchers. Moreover, climate change and environmental effects of carbon emissions were also significant points of concern in carbon emission research. The key findings of this study will be beneficial for the policymakers, academics, and institutions to determine the future research directions as well as to identify with whom they can consult to assist in developing carbon emission control policies and future carbon reduction targets.

**Keywords:** carbon emissions; scientometric; research trends; China; global research

---

## 1. Introduction

Climate change is one of the most prominent global issues that has attracted the attention of global academic researchers, policy makers and other related professionals. Climate change has caused several issues, such as global warming, ecological imbalance, technological issues, economic issues, and societal issues. Increasing concentration of greenhouse gas emissions is considered as a prime cause for these issues [1]. Thus, greenhouse gas reduction has become a key agenda of the global community. The Kyoto Protocol specified six main greenhouse gases which significantly impact the environment: namely $CO_2$ (carbon dioxide), $CH_4$ (methane), $N_2O$ (nitrous oxide), HFCs (hydrofluorocarbons), PFCs (perfluorocarbons) and SF6 (sulfur hexafluoride). Amongst them, carbon dioxide has been considered as the most prominent contributor to global climate change [2]. For instance, carbon dioxide emissions have increased from 22.15 Gt in 1990 to 36.14 Gt in 2014 [3]. According to Heede [4], 80% of global

carbon emissions are caused by urban human activities. Human activities such as fuel combustion during vehicular transportation, power generation emits large quantities of carbon dioxide to the environment. Moreover, construction operations and other industrial operations have also been recognized as major carbon emission sources. Thus, global researchers have been significantly focused on investigating methods to reduce carbon emissions. Accordingly, carbon emission monitoring at different levels (product, organization, city, and national) has been recognized as an important reference in driving the environmental strategies and policies towards carbon emission mitigation.

As a result of this increasing attention towards global carbon emissions, researchers have begun quantifying the carbon emissions [5]. While some of these studies focused on quantifying the carbon emissions on a national scale, some studies have quantified the carbon emissions on a global scale. For example Wang et al. [6] applied decomposition method to decompose aggregate environmental indicators, while Voigt et al. [7] studied the energy intensity. Moreover, Lan et al. [8] explored the energy footprint and Xu et al. [9] investigated the industrial carbon emissions. Global scaled carbon emission quantification studies revealed China as the largest carbon emitter since 2008 [2,10]. According to Zhang and Da [2], China accounts for 27% of the global carbon emissions and as indicated by Michaelowa and Michaelowa [11], emission value of China has increased by 174% during 1990–2010, while it has been doubled during 2001–2010. Moreover, these carbon exploration studies identified countries such as the USA, UK, and Australia as significant contributors to global carbon emissions. Therefore, it is evident that all the countries are responsible for the global carbon emissions in different scales. Carbon emissions of a country are influenced not only by its development demand but also by the production demand of other countries [12]. Therefore, decision makers across the globe have relied on research outcomes to understand the current situation of carbon emissions and derive future plans to reduce carbon emissions.

Over the past decade, researchers, industry practitioners, and various government authorities across the globe have given greater attention towards carbon emission reporting and identification of climate mitigation strategies, resulting in a significant rise in related research publications and works. As Hammond and Norman [13] indicated, analyzing past trends of carbon emissions have become a useful method to understand the current emissions and thereby predict future emissions. Accordingly, several studies have been conducted to analyze the past trends of various industrial sectors which include manufacturing [14] and energy [15]. Moreover, studies such as Hong, et al. [16] and Lu, et al. [17] which explored carbon emissions of the construction industry have been conducted to gain an insight into the carbon emissions of different industries. Moreover, global researchers have predominantly focused on exploring operational carbon emissions which have resulted in a significant increase in embodied carbon (EC) emissions. It is predicted that if the proper focus is not given to EC emissions, it is likely to increase even further in the future [18]. This has resulted in researchers focusing on capturing EC emissions and conducting more related research studies. This indicates that a surge of focus on certain aspects of carbon emissions such as operational carbon emissions may result in neglecting another useful aspect such as EC emissions.

*Research Gap, Objective and Research Novelty*

The surge in the carbon emission research domain encompasses several risks of neglecting important areas of research, which occurs due to the inability to cover the status quo of the domain. It is, therefore, necessary to rigorously analyze the domain to solve this scientific question. Several micro level literature reviews on carbon emission-related literature are available such as the review by Guo, et al. [19] on emission rates of indoor ozone emission devices, the review by Bartolini, et al. [20] on Green Warehousing and the review by Evans, et al. [21] on black carbon emissions in Russia. However, a macro level review on the carbon emission-related literature is yet to be discovered in bibliometric databases. Although there have been previous research and review papers on carbon emissions and other related fields, they have focused mainly on manual review, scientific review without depth and broad perspectives, and the adoption of a single approach. Therefore, an important research gap has been found in this domain due to the unavailability of review studies related to global carbon emissions. When analyzing bibliometric databases, such as Web of Science and Scopus, it was found that research studies related to carbon emissions initiated

back in the early 1980s and increased gradually over time. Accordingly, analyzing available bibliographic data on the domain will enable the researchers to identify the potential areas that require more attention in dealing with carbon mitigation strategies. In addition, it will provide an insight into what can be learned from the available literature.

This paper aims to address this research gap through a scientometric review of research on global carbon emissions. The scientometric review adopts a quantitative methodology to analyze the landscape and the intellectual core of the existing body of literature available on carbon emissions. This paper will also examine the deficiencies of the existing body of carbon emission knowledge by identifying and discussing the quality and the scope of available publications. The research findings provide a detailed understanding of the current state of global carbon emission research work and potential directions for future research. The findings will also be an updated and valuable reference for the practitioners and policymakers in planning their future work and funding.

Figure 1 indicates the approach followed in achieving the objectives mentioned above. A large corpus of bibliometric records (2945 journal articles) is analyzed, which is a significantly larger volume of articles than any study on carbon emissions has previously taken into consideration. Section 3 of this paper discusses some available literature on carbon emissions to assess the current situation of the research domain. Section 3 explains the approach utilized for literature search and indexing strategy as well as the scientometric techniques used for the analysis. While Section 4 analyses the bibliometric records using the scientometric techniques, Section 5 provides a conclusion based on the findings. The findings of the study are expected to contribute to the existing body of knowledge by highlighting the patterns and trends of carbon emission research. Moreover, it will map the network of key researchers on the domain as well as the key institutions and it will also recommend areas for future research studies.

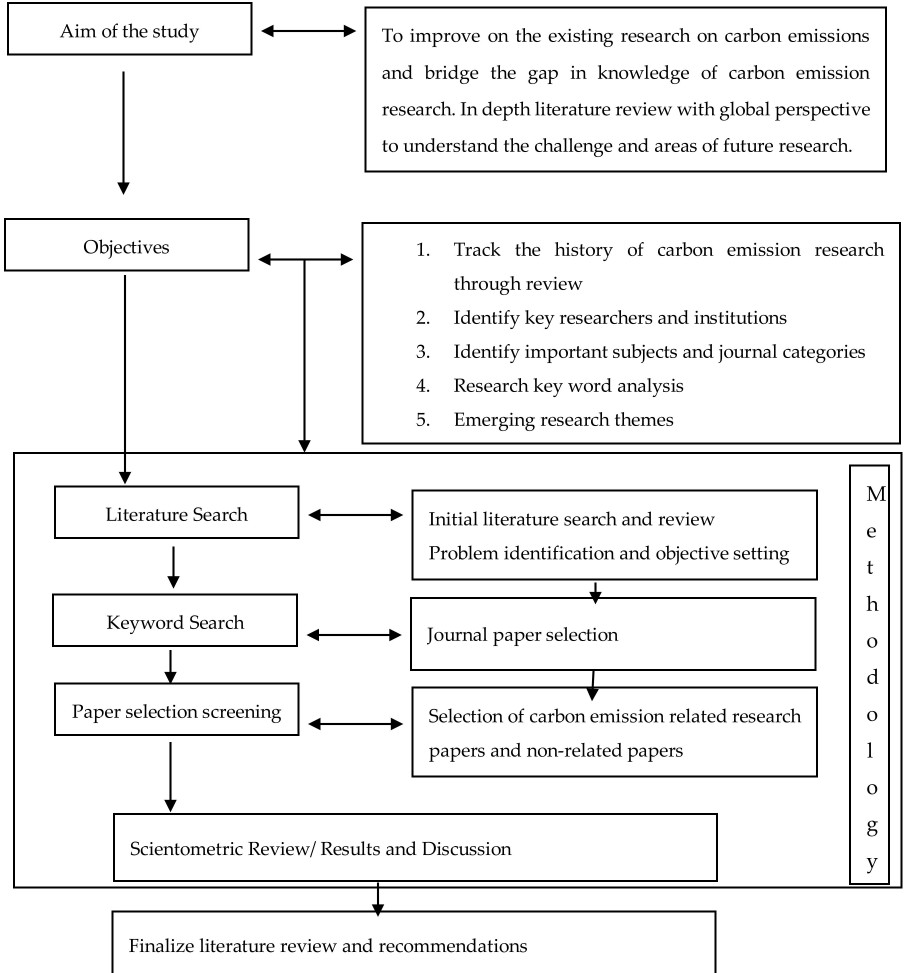

**Figure 1.** Research gap, methodology.

Considering these identified gaps in knowledge with a dearth of in-depth literature review on carbon emission research, this study is a timely improvement on the existing studies and bridge the literature gap in knowledge. It improves previous studies of carbon emissions with a robust review on a global perspective and identified areas of further research. The study aim is to provide literature support for carbon emission research using in-depth scientometric review software (CiteSpace) to ensure that researchers and professionals have a comprehensive understanding of the research trends and future direction for management strategies in carbon emissions. The objective and novelty of this study, as illustrated in Figure 1, is based on the research gap and the aim identified in this research domain. The novelty and objectives include the five scientometric analysis areas investigated in the study: (i) track the history of carbon emission related research, (ii) identify the key researchers and institutions, (iii) identify important subjects and journal categories, (iv) research keywords and (v) timeline representative with emerging research themes. Figure 1 also describes the research overview and strategy adopted in the study from the research aim to methodology and recommendations. This figure also helps to explain and illustrate a graphical overview of the research gap, objectives and the research novelty.

## 2. Literature Overview

Carbon emission research was initiated in 1981 with the first publication focusing on volatile organic carbon emissions of cooling tower water [22]. Thereafter several researchers explored the domain without significant impacts until the Kyoto protocol was signed in 1997. However, carbon

emission research became a trending topic after 2007, resulting in a significant increase in research publications. With the rising global temperatures, climate change has become a major global concern which is also considered as the most serious issue global community has to address in the 21st century [23]. Research on global carbon emissions has significantly increased after discovering carbon emissions as the major cause of climate change.

Increasing carbon emissions have caused significant concern amongst the countries such as China, United States, Russia, India, European Union, and Japan as the leading carbon emitters of the world [24]. Carbon emission research expands over several research areas which include environmental sciences, engineering, economics, energy, etc. Table 1 indicates the top 10 research areas which carbon emission research has impacted.

**Table 1.** Top 10 research areas affected by global carbon emission research.

| Category | Record Count | Percentage |
|---|---|---|
| Environmental Sciences | 1374 | 46.66% |
| Green Sustainable Science Technology | 635 | 21.56% |
| Engineering Environmental | 586 | 19.90% |
| Environmental Studies | 557 | 18.91% |
| Energy Fuels | 535 | 18.17% |
| Economics | 367 | 12.46% |
| Engineering Chemical | 143 | 5.81% |
| Meteorology Atmospheric Sciences | 157 | 5.33% |
| Ecology | 104 | 3.53% |
| Engineering Civil | 85 | 2.89% |

Accordingly, it is evident that the majority of carbon emission research relates to environment-related aspects. Economics, energy fuels, ecology, and civil engineering are some of the other notable research areas which have been affected by carbon emission research. A major reason for carbon emission research to expand into the aforementioned research areas is the goal of sustainable development which has become increasingly popular among the global community due to low carbon emission society concept.

Initially, carbon emission research focused on capturing global emission trends and practices. However, with the heightened significance of sustainable development, researchers have focused on developing methods to comprehend and mitigate the effects of carbon emissions [25]. Life Cycle Assessment is recognized as one of the basic methodologies which consider the inputs and outputs associated with a process or a product in determining the carbon emissions [26]. As sustainable development required the integration of not only environmental but also social and economic components, carbon emission research expanded over various research areas. Despite various methods and concepts developed for social and economic evaluations of carbon emissions such as Social LCA (SLCA) [27] and Life Cycle Cost Analysis (LCCA), researchers have kept searching for new opportunities and developing new methods such as carbon emission trading, carbon tax, etc. [28–30].

The level of vulnerability to the effects of climate change and impact created by climate change are different from one country to another. Their capacity to address mitigation and adaptation also vary significantly. Accordingly, carbon emissions are monitored at different levels (product, organization, city and national) which enables them to drive carbon mitigation strategies and implement policies to reduce carbon emissions. In the recent past, there has been significant interest in revealing and estimating carbon emissions at these levels.

In the last decades, there has been a growing interest in estimating and revealing carbon emission drivers via carbon footprint analysis at different scales. The carbon footprint originates from the concept of ecological footprint which is a measure of the impact on the environment expressed as the amount of land required to sustain natural resources. However, the carbon footprint of a functional unit, when it is not associated with the ecological footprint, is the climate impact under a specified metric

that considers all relevant emission sources, sinks and storage in both consumption and production within the specified spatial and temporal system boundary [31]. Although a specific definition of carbon footprint has been not stated, according to Wiedmann and Minx [32] carbon footprint is the measure of the total amount of carbon dioxide emissions directly and indirectly caused by an activity or accumulated over the life stages of a product. Specifically, the carbon footprint is the overall amount of carbon emissions associated with a country, city or product, along its supply chain including end-of-life recovery and disposal. The carbon footprint is an environmental indicator and, as such, it needs to be used in appropriate contexts thus providing the right information. When properly used, it is essential for making decisions and performance evaluations and for allowing policymakers to have a solid basis upon which climate policies can be established and implemented.

Several carbon emission research studies have grabbed the attention of global researchers due to the rapidly changing global climate. Among the most cited research articles available in Web of Science research database, a study conducted by Allen, et al. [33] on "warming caused by cumulative carbon emissions towards the trillionth tonne" was ranked as the most cited article. The article, which was cited 632 instances by researchers, revealed that anthropogenic emissions of one trillion tonnes of carbon are likely to cause an increase of global temperatures by two degrees Celsius. As a result of comprehensive calculations and policy considerations, this research article is highly recognized among global carbon researchers. The research article by West and Marland [34] was the second most cited article. This study focused on agriculture-related carbon emissions and calculated the impact of the US agriculture sector towards global carbon emissions. The study further indicated that adopting newer technologies and methods to harvest crops emit less carbon compared to conventional methods. This further indicates that global carbon researchers are in search of methods and technologies to reduce global carbon emissions.

The third most cited article in the WoS database discussed how household actions can contribute to reducing the US carbon emissions [35]. Authors revealed that national implementation of interventions on five distinct categories of household actions could reduce 20% of the US household carbon emissions. The research article on "energy consumption, carbon emissions and economic growth in China" by Zhang and Cheng [36] was cited 499 times and was ranked as the fourth most cited article on carbon emissions available in the WoS database. This study explored the carbon emissions of China from 1960 to 2007. "Energy consumption, income and carbon emissions in the United States", which was cited 434 times by global researchers, by Soytas, et al. [37] was next on the list. Accordingly, large countries such as China and the United States are significantly concerned about the impact of global carbon emissions to their economy and have taken significant efforts to investigate. Both studies mentioned above revealed that income growth is not a solution to environmental problems.

These studies clearly indicate that global researchers are focusing heavily on different areas of the carbon research domain. Thus, exploring the patterns and trends of global carbon emission research would provide a thorough insight into the carbon emission research domain.

## 3. Methodology

Science mapping, which is defined as "a generic process of domain analysis and visualization", can detect the intellectual structure of a scientific domain [38]. This method is useful in visualizing trends and patterns of a large body of bibliometric data, allowing researchers to make discoveries related to a particular scientific domain. Several scientific methods, such as content analysis [39], latent semantic analysis (LSA) [40], literature reviews [41], bibliometric techniques [42–44] and scientometric analysis [45] have been used by researchers in different research areas, such as building information modelling (BIM), green building and innovation and energy and sustainability. However, a scientometric analysis provides several advantages over the other analytical techniques by providing a broader approach to identify the insightful patterns and trends of a large bibliometric dataset [46]. This study, therefore, adopted a scientometric analysis technique to analyze the large body of literature

available on global carbon emissions. Accordingly, the research methodology consisted of the selection of science mapping tool, data collection, data analysis, visualization, and interpretation of the results.

This study adopts a scientific and visual representation (illustrative diagrams and maps) with a qualitative (descriptive) research approach for global literature review [47]. To explore the characteristics and productivity of global research with the research objective of providing an in-depth review of carbon emission research [48,49]. The descriptive and visual representation was adopted considering its ability to analyze and discuss academic research within the entirety of scientific purview comprehensively [50–52]. Scientometric review with descriptive and visual illustration is described as one of the most adopted methods of evaluating and assessing research trends by researchers, academics, institutions, countries, and journals. As adopted in this study, it helps to understand the structure (cluster), research areas and trends in carbon emission better than [53,54] and other studies.

### 3.1. Selection of Science Mapping Tool

There are several science mapping tools available to analyze insightful patterns and trends of a scientific domain. These tools possess their own pros and cons when performing a scientific analysis. It is therefore important to select the best suitable tool to analyze a scientific domain. Among various tools available, CiteSpace and VOSviewer were identified as the two most suitable tools to perform this analysis. VOSviewer provides basic software tools for exploring, analyzing and visualizing bibliometric data. CiteSpace offers more visualized analytical options, providing more opportunities to address important elements of a research domain, such as links between publications and major components of the domain. Compared to VOSviewer, CiteSpace has the ability to analyze a larger bibliographic dataset. Accordingly, CiteSpace was selected as the science-mapping tool for this research.

### 3.2. Data Collection

The selection of a scientific database is a critical step in a review study as there are multiple scientific databases, such as Scopus, Web of Science (WoS) and Google Scholar. Apart from these major databases, core journal publishers also possess their own databases, such as Emerald, Elsevier-Science Direct, Wiley Online, Taylor and Francis, IEEE Explore and Springer Link. However, following the previous researchers [42,49,55], the Web of Science core collection database was selected for this review. The WoS core collection consists of the most influential and related journal records and combined with WoS scientific robustness, this database provides a wider range of bibliographic data to the user [42,51]. Moreover, CiteSpace enriches with several data sources, such as WoS, Scopus, and PubMed, and converts those data, from different sources to WoS format prior to processing. Thus, to prevent losing data from the conversion process, the WoS core database collection was selected as the source of data for this review.

The data source and retrieval strategy adopted was based on the best techniques that will achieve the established aim, objectives and the identified methodology of this study. The source of data includes journal articles published from 1980 to June 2019. Journal papers are significant and reliable as the primary source of academic research literature review [56,57] and it has been adopted for similar purpose by studies such as [58,59]. June 2019 was adopted to avoid the error of hasty generalization because several relevant research can still be published in the year 2019. However, few relevant referred journal articles in the year 2019 were adopted and referenced as used in the traditional (manual) literature search and retrieval techniques.

A comprehensive bibliographic search, extraction and indexing were carried out on WoS core collection using the search string "Greenhouse Gas Emissions*", "Carbon Emissions" and "Carbon footprint". The time span of the publications retrieved ranges from 1980 to June 2019 (29 years). The search results were refined only to include journal articles and articles written in English. Due to the thorough peer review process involved, journal articles are considered more trustworthy compared to other sources [42]. Moreover, most authors usually republish their conference papers in scholarly journals, which is considered the "certified knowledge" [60]. Accordingly, a total of 2945 journal

articles were retrieved in June 2019 and the records were downloaded as text files containing the full record and cited references. The very first paper on carbon emissions has been published in 1981, which focused on developing a tool to measure carbon emissions of cooling towers [22]. Subsequently, there have only been very few publications on the domain until 2010. Thereafter, there has been a significant increase in publications and 2018 recorded the highest number of publications (479).

This review data search and retrieval approach identifies the trend in carbon emission research using the traditional search of related articles to extract the relevant discussion over a period. This technique includes the search of the research topics, keywords from a search engine such as Ask.com, AOL, Bing, Google and Yahoo [46]. This approach does not only provide the preliminary insight into the research area, but it also helps to verify the validity and reliability of the bibliometric approach used. It helps to create a foresight of what to expect in the scientometric analysis (qualitative and visualization) and consequently make significant relevant discussion.

### 3.3. Data Analysis Strategy

Scientometric analysis strategies adopted in this study include (1) tracking the history of carbon emission through literature search and extraction; (2) bibliometric search to identify the key researchers and institutions. The important subjects and journal categories, research keywords, representative references as well as relevant salient and emerging research themes; (3) science mapping through CiteSpace to visualize the results of the analysis; and (4) make a qualitative discussion to achieve the objective of the research. This strategy was adopted based on its validity by exploring several research studies which include Liang, et al. [61], Jin, et al. [62] and Zhao [51].

### 3.4. Scientometric Analysis

Several scientometric techniques have been identified by [40], such as author co-citation analysis, keyword co-occurrence analysis, and document co-citation analysis. Initially, networks were constructed through keyword co-occurrence analysis, direct citation analysis, co-authorship analysis, and citation burst analysis. This was followed by an analysis of useful information of the domain, which displays "the conceptual, intellectual, or social evolution of the research field, discovering patterns, trends, seasonality, and outliers" [46]. A scientometric analysis is a quantitative study of science and scientific communication in research or policy statement [51,63,64]. Scientometric analysis as adopted in this study includes measuring the impact of communication links, referencing sets of articles, countries, journal, institution impact and scientific citations among others [65–67]. The analytical approach investigates a wide range of carbon emission subject areas for scientific knowledge mapping. It identifies and captures knowledge area, structural patterns, trace the history and include prominent or salient relevant ideas for research decision. The scientometric analysis is adopted as a scientific process of analyzing research in this study because it uses mathematical formulae and visualization techniques in a systematic order as preferred by the researcher. Also, this scientific method has been adopted in several ways for capturing research knowledge such as bibliometric techniques [43,44,60]; content analysis [39]; literature review [41,68,69]; latent semantic analysis (LSA) [40,51]; and scientometric visualization with different software [45,70].

CiteSpace is a Java software adapted to process article data obtained from Scopus database for scientific information and visualization, mining and analyzing data for scientific uses [52,71–73]. It is used for several scientific knowledge networks and gathering information involving cited references, authors, co-occurring, bibliographic sources, keywords and many more. CiteSpace network analysis includes 'nodes' which show the result of significant collections of data such as keywords, authors, documents, institution, and region (countries) for making research inferences. The node size indicates the article or journal frequency, citation count or impact of the subject study. For example, the larger nodes reflect higher frequency or citation counts, and the combination of these nodes into groups is referred to as cluster(s), and it signifies an important domain or intensity of a specific thematic. While the links between two entities (authors, references) represent how frequently the two entities are cited together by other entities, the betweenness centrality result shows the impact of a node on

another node. Similarly, the larger the centrality, the higher the impact of the entity on the other entities with the tendencies of becoming a key entity.

The analysis conducted in this study includes significant cited articles, articles that attract research community attention, keywords with strong frequency surge to identify emerging topics and areas of further research in carbon emission research. The analysis involves six basic stages, namely the (1) Scopus bibliographic database search, (2) literature search indexing, (3) scientometric analysis, (4) the use of CiteSpace software, (5) the result and discussion in two subsections (trends of carbon emission related global research analysis and future research areas of carbon emission research), and (6) the conclusion. The first stage described how the data are accessed from the Scopus bibliographic database search. The second stage involves the sorting and indexes applied to retrieved articles from the first stage. While the third stage is the software (CiteSpace) which accept the RIS data format and process it for the required types of analysis in the study, the fourth stage is the analysis conducted using the CiteSpace software, and this is adopted for the result discussion and conclusion of this study.

## 4. Results and Discussion

### 4.1. Global Trends of Carbon Emission Research

The first research publication on carbon emission was in 1981, and since then carbon emission research has gained ground due to increased global warming. Figure 2 depicts the gradual increase of research publications related to the domain starting from 1980 to June 2019. In 2018, there have been 479 research articles related to GHG emissions, recording the highest number of publications on the domain. According to Figure 3, carbon emission research has increased significantly since 2007. The highest number of publications on carbon emission research was reported in 2018—479 publications. There were 220 publications reported already by June 2019, implying a significant increase in publications on this domain in 2019. The carbon emission research domain remains to be a key theme of sustainability research and many researchers and institutions, therefore, tend to explore the domain extensively. In order to explore the research patterns of carbon emission research, a co-authorship analysis, country, and institutional analysis and a co-citation analysis were conducted.

The increasing trend of carbon emission research articles indicates the enhanced attention towards this domain of the researchers. This is a further indication that carbon emission research domain has become a highly popular domain amongst the global researchers. Rapidly changing global climate change can be recognized as one of the major reasons for this increased level of attention. Moreover, the attention of global leaders and policymakers to identify possible climate change mitigation strategies is another reason for this surge in carbon emission research.

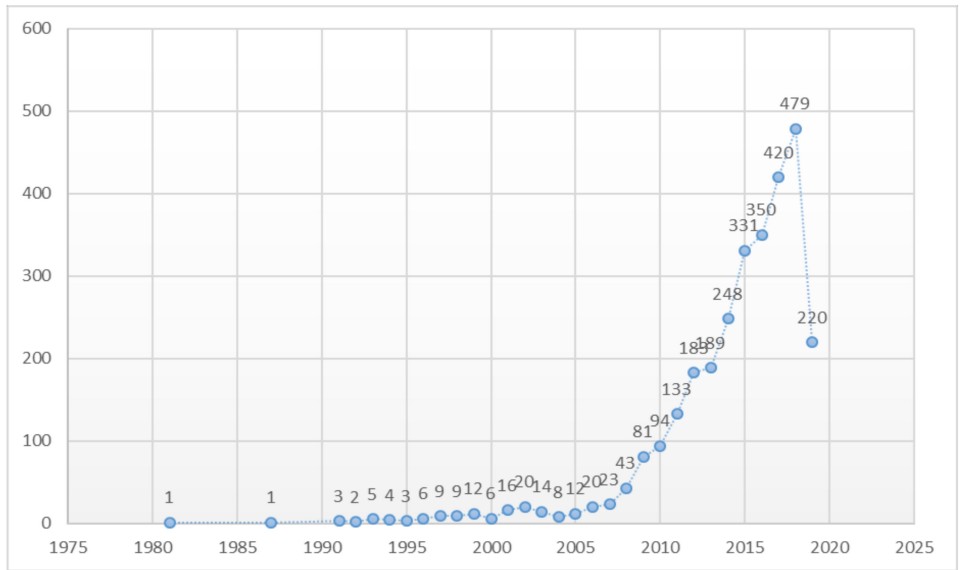

**Figure 2.** The trend in carbon emission publications from 1981 to June 2019.

### 4.2. Co-Author Scientometric Analysis Rresults

The co-author scientometric analysis results examined and discussed the trends of authors participating in carbon emission research. The author's impacts are measured by citation, institution, and the network distribution of authors by countries. This author analysis is essential to understand the historical trends of carbon emission research by the researcher. This analysis is essential to determine the researcher with whose work must be given adequate consideration to support/corroborate new findings or refute it and/or make a comparative study as necessary. The co-authorship network is displayed in Figure 4, in which nodes represent the authors and the links showing the collaborations between authors. The network pruning was used to remove the excessive links. Accordingly, 548 nodes and 602 links were identified in the co-authorship network. As illustrated in Figure 3, a significant number of links between the nodes indicate that authors have an extensive collaboration network. Thus, it is evident that authors are heavily relying on collaborations to come up with significant outcomes in the research domain.

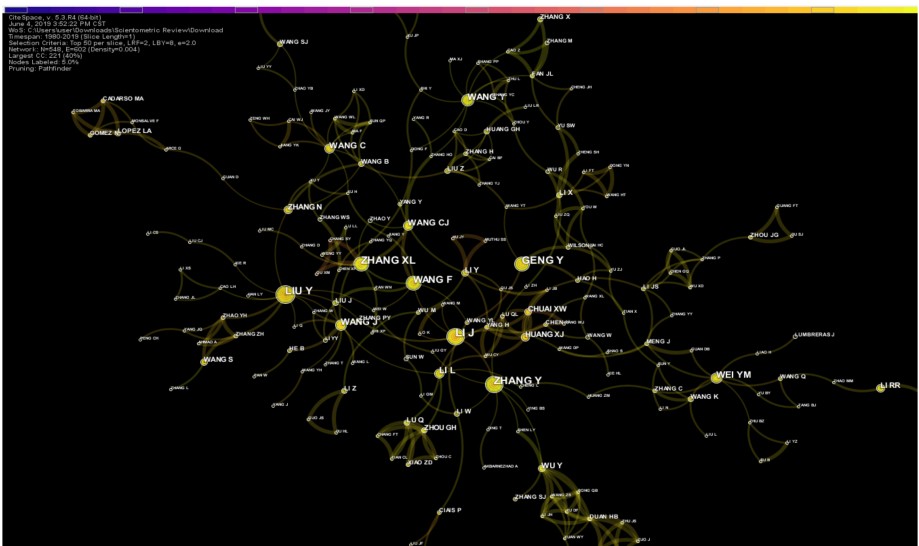

**Figure 3.** Co-author scientometric analysis by research count and cluster impact.

As illustrated in Figure 4, top-ranked author by publication count is Li (2011), with a count of 19, followed by Liu (2014), with a count of 17. Zhang Y. (2014) is the third ranked author, with a count of 16, Geng (2013) followed with a count of 15, and fifth is Zhang X.L. (2014), with a count of 15. Wang F. (2015), with a count of 14 takes the sixth position while Wang J. (2015), with a count of 12 is in the seventh position. Wang Y. (2018) with a count of 11 and Wei Y.M. (2016), with a count of 11 takes the eighth and ninth position respectively.

When analyzing the co-authorship network further, two authors with citation bursts were revealed. Considering the authors' burst in co-author and clusters of similar research interest, the top author is Matthews HS (2004), with a burst strength of 4.67, followed by Flysjo A (1980), with a burst strength of 3.76. The co-author and cluster analysis results, as illustrated in Figure 4 is corroborated with Table 2 scientometric analysis of the top 15 most productive authors on carbon emission research. The results elaborated in both Figure 3 and Table 2 are closely related, and it identified 13 authors whose work should be adequately considered.

**Table 2.** Co-author analysis of most productive authors on carbon emission research.

| Year | Author | Institution | Country | Counts | H-index | Rank |
|------|--------|-------------|---------|--------|---------|------|
| 2011 | Li J. | Zhejiang Gongshang University | China | 19 | 8 | 1 |
| 2014 | Liu Y. | Chinese Academy of Sciences | China | 17 | 60 | 8 |
| 2004 | Zhang Y. | Beijing Normal University | China | 16 | 32 | 4 |
| 2014 | Geng Y. | Shanghai Jiao Tong University | China | 15 | 52 | 9 |
| 2010 | Zhang X.L. | Nanjing Normal University | China | 15 | 6 | 6 |
| 2015 | Wang F. | China University of Mining and Technology | China | 14 | 4 | 3 |
| 2015 | Wang J. | China Building Materials Academy | China | 12 | 9 | 10 |
| 2018 | Wang Y. | Dongbei University of Finance and Economics | China | 11 | 5 | 2 |
| 2016 | Wei Y.M. | Beijing Institute of Technology | China | 11 | 50 | 5 |
| 2015 | Li L. | Tsinghua University | China | 10 | 27 | 11 |
| 2015 | Wang C. | Beijing Institute of Technology | China | 9 | 6 | 7 |
| 2015 | Wang C.J. | Guangzhou Institute of Geography | China | 9 | 11 | 12 |
| 2012 | Huang X.J. | Nanjing University | China | 8 | 1 | |

All 13 authors indicated in Table 2 represented China-based universities which was the notable fact. This indicates that China is a major contributor to carbon emission research domain. Impact of China and China-based institutions will also be explored in the upcoming sections of this review. Co-authorship network was further analyzed using content analysis to gain an insight into the work of the most prolific authors related to carbon emissions.

Li J. is the most prolific author in terms of publication count and when exploring his publications, it was revealed that his initial publication was done in 2014. Li J. has mainly focused on transportation-related carbon emissions, built environment and energy-related carbon emissions. Out of the 19 publications of Li J., the most cited paper is titled "Decoupling urban transport from GHG emissions in Indian cities— A critical review and perspectives" [74]. This study was focused on capturing the GHG emissions of the public transportation systems in India and thereby developing a more efficient system to mitigate GHG emissions. This study grabs more attention as it is related to a developing country and Li J. has managed to explore a vital aspect of a developing country, the transportation systems. Second most cited publication of Li J. explored the life cycle carbon emissions of building maintenance using a case study in Hong Kong [75]. The most reason publications of Li J. focused on the household carbon emissions of China [76] and explored the spatial correlation network structure of carbon emissions using social network analysis (SNA) [77]. These studies clearly indicate that Li J has explored different dimensions of carbon emissions through his research.

Liu Y. is the second most prolific author in terms of publication count, and in terms of the H-index, Liu Y. is the top-ranked author among the 13 authors indicated in Table 2. Aggregate and disaggregate

based analysis of the Indian economy on carbon emissions and energy consumption [78] is the most cited work of the author. Once again this indicates that global researchers are more concerned and attracted to carbon emission related research on developing countries. Moreover, due to the scarcity of research work on developing countries, available literature has been highly cited by the authors worldwide. The second most cited paper of Liu Y. has addressed carbon emissions of China using a new concept, a spatial Durbin model [79]. Thus, it has managed to grab the attention of the researchers which has made it one of the highly cited papers of the author. Similarly, the study on " driving factors of carbon emissions embodied in China-US trade: a structural decomposition analysis" [80] was amongst the highly cited papers of the author due to the novelty of his research approach. It was evident that Liu Y. has used different approaches to capture carbon emissions which has resulted in making him the most prolific author.

Geng is the second most prolific author in terms of the H-index in the carbon emission research domain with an H-index of 52 and fourth in terms of the research publication count. Geng has investigated several aspects of carbon emissions, which includes a study on carbon reduction through industrial symbiosis, using a cement production case study [81], a calculation methodology of carbon emissions of urban energy consumption [82], and several studies on carbon emissions of urban cities [12,83,84]. The study on identifying drivers and barriers for reducing carbon emissions of supply chain system among Chinese manufacturers [85] was the most cited paper by Geng in this domain. Thus, it is evident that Geng is mostly focused on exploring carbon emissions of the construction and built environment as well as the urban cities. This further indicates that global researchers are significantly concerned about evaluating the impact of the construction industry and urban cities on global carbon emissions. Wei is ranked third on this list with an H-index of 50. When exploring the publications by Wei, the study on carbon emission patterns in different income countries [86] was the first publication on carbon emission research. The study on the impact of lifestyle on energy use and $CO_2$ emission, titled 'An empirical analysis of China's residents' [87] was one of the highest cited publications by Wei in this domain. Compared to the other authors, Wei has conducted more studies in China to explore carbon emissions.

Institutions have a significant impact on the research domain. Therefore, adopting the co-authors by the institution to understand the institutions' rank or department with a high concentration of carbon emission research is another useful aspect by which to explore the trends and patterns of the research domain. As indicated in Figure 4, the institutional network of co-authors included 294 nodes and 317 links. Accordingly, results revealed the Chinese Academy of Sciences (2010) in cluster #0 as the top-ranked institution by centrality, with a centrality of 0.27. Moreover, Chinese Academy of Sciences indicated 72 records which are at the top in terms of frequency as well. Tsinghua University (2010) in cluster #0, with a centrality of 0.16 and a frequency of 34, is in second place among the institutions. While University of Florida (2010) in cluster #3 with the centrality of 0.12 and a frequency of 10 is in third in the list, University of Leeds (2013) in cluster #1 with a centrality of 0.11 and frequency of 15 is the fourth. The Hong Kong Polytechnic University (2011) in cluster #8 with a centrality of 0.09 is fifth, University of East Anglia (2012) in cluster #1, University of Cambridge (2010) in cluster #1 and Aarhus University (2011) in cluster #7 with the centrality of 0.06 is in equal sixth position of the list. Next in the list is University of Sydney (2009) in cluster #2, the University of Maryland in cluster #1, US Forest Service (2015) in cluster #4 and Harbin Institute of technology (2008) in cluster #8 with a centrality of 0.05. Table 3 indicates the countries which the above-indicated universities are located. This provides a basic insight into the country related contributions to the carbon emission research domain.

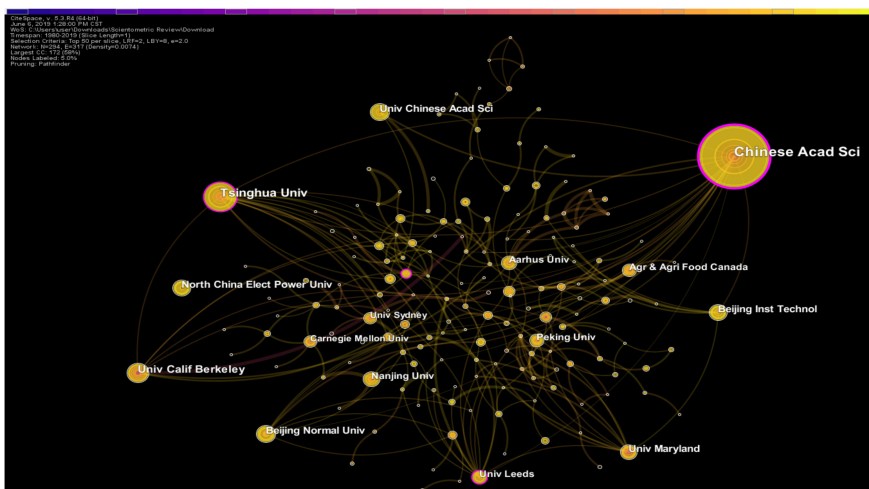

**Figure 4.** Co-author scientometric analysis of carbon emission research by the institution.

**Table 3.** Institutional analysis of most productive authors on carbon emission research.

| Year | Institution | Country | Centrality | Frequency | Cluster # |
|------|-------------|---------|-----------|-----------|-----------|
| 2010 | Chinese Academy of Sciences | China | 0.27 | 72 | 0 |
| 2010 | Tsinghua University | China | 0.16 | 34 | 0 |
| 2010 | University of Florida | USA | 0.12 | 10 | 3 |
| 2013 | University of Leeds | UK | 0.11 | 15 | 1 |
| 2011 | Hong Kong Polytechnic University | Hong Kong | 0.09 | 13 | 8 |
| 2012 | University of East Anglia | UK | 0.06 | 4 | 1 |
| 2010 | University of Cambridge | UK | 0.06 | 10 | 1 |
| 2011 | Aarhus University | Denmark | 0.06 | 16 | 7 |
| 2009 | University of Sydney | Australia | 0.05 | 14 | 2 |
| 2002 | University of Maryland | USA | 0.05 | 19 | 1 |
| 2015 | US Forest Service | USA | 0.05 | 5 | 4 |
| 2014 | Harbin Institute of Technology | China | 0.05 | 7 | 8 |
| 2011 | Beijing Normal University | China | 0.05 | 22 | 0 |

According to Table 3, it is evident that the majority of the institutions are located in China, which is followed closely by the USA and UK. Thus, China, USA, and the UK can be considered as three countries with a significant impact on carbon emission research. However, the impact of each country on the global research on carbon emissions will be explored further in the upcoming sections of this paper. Moreover, this network analysis revealed two institutions with citation bursts. A citation burst indicates a prolific duration of publications of a certain research domain. These bursts can either be from a country, institution or an author. Figure 5 indicates the citation bursts of the two institutions, the University of Sydney and North China Electric Power University. The University of Sydney recorded the citation burst in 2009 which expanded up to 2011 and citation burst of North China Electric Power University was a more recent burst. As per the findings above, the majority of the most productive institutions on carbon emission research domain was reported from China, USA, and the UK. Therefore, to analyze the impact of different countries to the domain, a country network analysis was conducted.

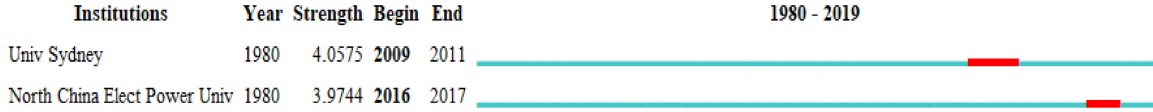

**Figure 5.** Citation bursts of the institutional network.

As indicated in Figure 6, co-author network analysis revealed Peoples Republic of China (2005) in cluster #1 with a citation count of 884 was ranked in the first place. The second is USA (1993) in cluster #0 with a citation count of 674, third is United Kingdom (1999) in cluster #2 with a citation count of 233, and the fourth in Australia (2007) in cluster #4 with a citation count of 161. The fifth is Spain (2009) in cluster #3, with a citation count of 146, while equal in sixth is Canada (2001) in cluster #1 and Germany (2008) in cluster #0 with citation counts of 140. Ranked in eighth is Italy (2009) in cluster #1, with citation counts of 116. The tenth is The Netherlands (2001) in cluster #2, with citation counts of 96.

The top-ranked country by centrality is USA (1993) in cluster #0, with the centrality of 0.23, followed by The Netherlands (2001) in cluster #2 with the centrality of 0.18. The third is the United Kingdom (1999) in cluster #2 with the centrality of 0.17. Tied in fourth position in Spain (2009) in cluster #3, and France (2001) in cluster #0 with the centrality of 0.17. Australia (2007) in cluster #4 and Germany (2008) in cluster #0 are equally on sixth with the centrality of 0.15. Peoples Republic of China (2005), Italy (2002) in cluster #1, Finland (2011) in cluster #4 and Saudi Arabia (2013) are equal with the centrality of 0.06.

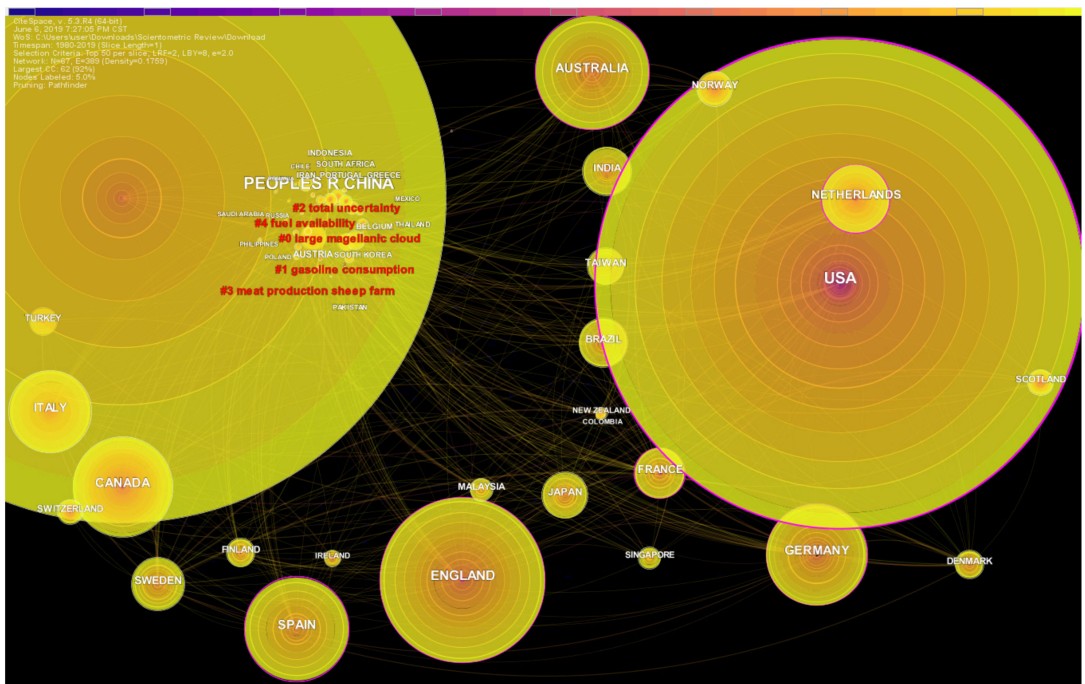

**Figure 6.** Co-author analysis of carbon emission research by countries.

Co-author network analysis revealed six significant citation bursts spanning over the considered period. These bursts indicate that these countries have been significantly involved in carbon emission research. It is noteworthy that the USA has the strongest citation burst, which spanned for 15 years from 1993 to 2008 recording the longest citation burst. The United Kingdom has had a nearly similar citation burst which spanned for 13 years from 1999 to 2012. The most recent citation burst on carbon emission research was recorded in 2016–2017 in Iran, which indicates that countries such as Iran are significantly interested in the carbon emission research domain.

The result of the co-author scientometric analysis revealed that Li J. and Liu Y. followed by Zhang Y. and Geng Y. are the most significant authors in carbon emission research with substantial article authorship. It shows how their co-author, burst analysis, centrality, H-index, inventions, and research approach has transformed the study of carbon emission research over the years. Meanwhile the history of the subject study from 1980-2019 also shows that the Chinese Academy of Sciences, Tsinghua University, University of Florida and the University of Leeds have contributed enormously

to the development of carbon emission research. Moreover, several countries such as the USA, United Kingdom, China, Italy, Spain, Australia, France, and The Netherlands have contributed significantly to the carbon emission research domain. As indicated in Figure 7, several countries also recorded citations bursts indicating a significant contribution to the carbon research domain. This further reflects that more is expected of these countries to pioneer the transformation of carbon emission research and contribute greatly by adopting and improving their invention or collaborative research.

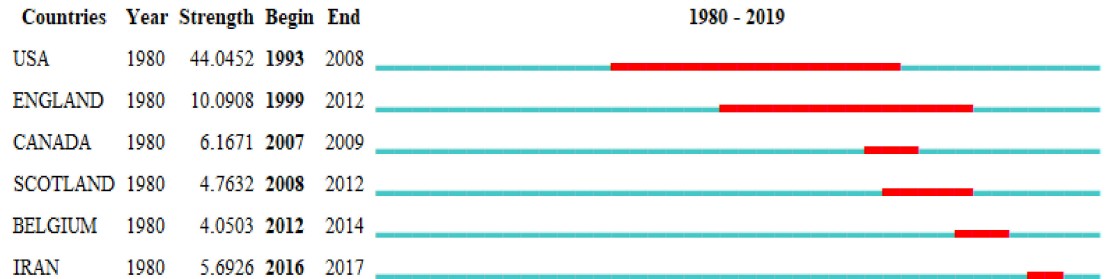

**Figure 7.** Citation bursts in the country network.

These findings imply that for significant authorship and co-authorship of research on carbon emissions, the studies of these significant researchers and countries shall be given adequate consideration either as the theory, principles and knowledge to corroborate or refute future invention, or authors, institutions and countries to collaborate with regarding research and applications or as studies and countries for comparative studies. For instance, a comparative study between theories, principles, methods, and applications towards carbon emission research improvements, and a comparative study between active carbon emission research countries and passive countries, with passive countries alone and vis-à-vis each other are significant implications of these results.

### 4.3. Co-Word Scientometric Analysis Result

The co-word scientometric analysis provides a scientific review of the keywords, and the association of words is often used in carbon emission research to understand the history and its relation to the global trends of carbon emission research. This section investigates and presents significant research keywords and co-words using CiteSpace software. The core contents of the research are presented by the keywords and indicate the development of research topics over a time period. Two types of major keywords are found in the WoS database: (i) keywords supplied by the authors and (ii) keywords identified by the journal which are also known as "keywords plus". Both types of keywords from the 2561 bibliographic records were utilized to build a network of co-occurring keywords which consists of 219 nodes and 816 links (Figure 8). The node size indicates the frequency of each keyword found through the analysis. While Figure 8 provides a broad visualization of the co-words with their network, Table 4 presents a detailed analysis of the top 20 co-words used in carbon emission research based on their frequency. The top ranked co-word by frequency is "carbon footprint" with (Freq. = 553) followed by "$CO_2$ emission" (503), "carbon footprint" (Freq. = 449), "greenhouse gas emissions" (Freq. = 445), "Sustainability" (Freq. = 98), "Concrete" (Freq. = 88), "Life cycle assessment" (Freq. = 83), "life cycle assessment" (Freq. = 404), "climate change" (Freq. = 337), "impact" (Freq. = 316), "energy" (Freq. = 291), "China" (Freq. = 276), "energy consumption" (Freq. = 272), "consumption" (Freq. = 217), "system" (Freq. = 210), "model" (Freq. = 202), "emission" (Freq. = 200), "economic growth" (175), "dioxide emission" (173) and "performance" (146). Observations of Figure 8 and Table 4 provide an interesting result that shows that the keywords ("carbon footprint" and "$CO_2$ emissions") with the highest counts by frequency also recorded the highest network.

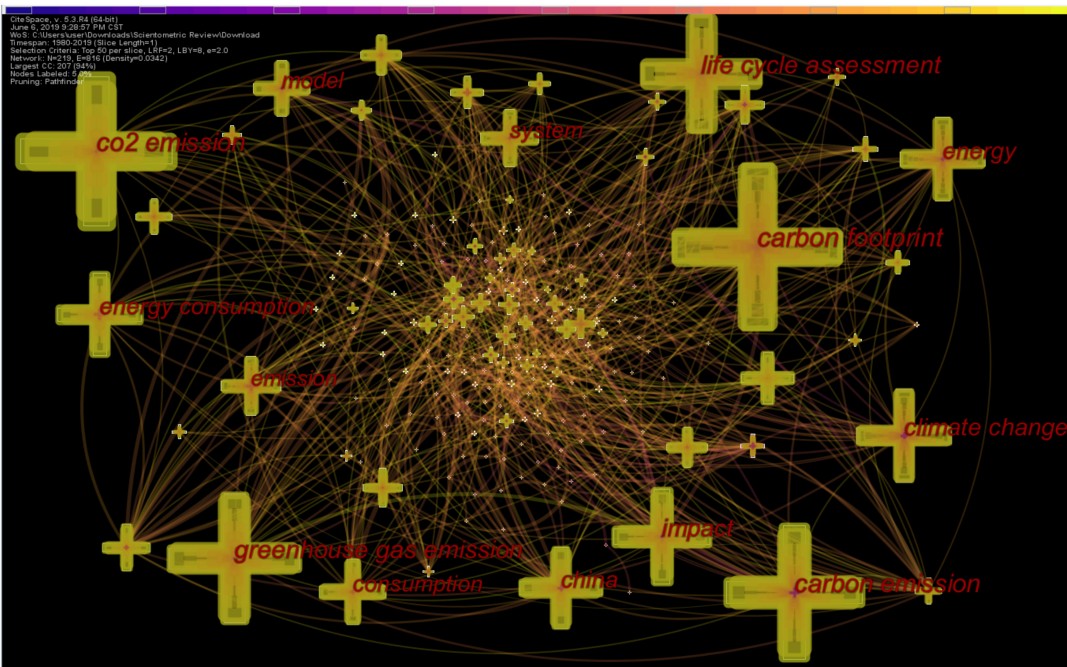

**Figure 8.** Co-word scientometric analysis of previous research on carbon emissions.

**Table 4.** Top 20 co-occurring keywords in carbon emission research according to the frequency.

| S/N | Freq. | Keyword | S/N | Freq. | Keywords |
|---|---|---|---|---|---|
| **1** | 553 | Carbon footprint | 11 | 217 | Energy consumption |
| **2** | 503 | $CO_2$ emission | 12 | 210 | consumption |
| **3** | 449 | Carbon emission | 13 | 202 | System |
| **4** | 445 | Greenhouse gas emission | 14 | 200 | Model |
| **5** | 404 | Life cycle assessment | 15 | 175 | Emission |
| **6** | 337 | Climate change | 16 | 173 | Economic growth |
| **7** | 316 | Life cycle assessment | 17 | 146 | Dioxide emission |
| **8** | 291 | Impact | 18 | 143 | Performance |
| **9** | 276 | Energy | 19 | 143 | GHG emission |
| **10** | 272 | China | 20 | 139 | Management |

Moreover, the co-word analysis revealed several citation bursts. Analysis results revealed 45 co-word citation bursts. Figure 9 presents the top 10 co-word citation bursts identified through the scientometric analysis. The strongest citation burst was recorded from the term "climate change" which spanned over a period of 11 years starting from 1999 and lasting until 2010. These citation bursts indicate salient topics and themes of carbon emission research and how carbon emission research focuses have shifted over time. Especially with global climate change becoming a significant area of concern, researchers are exploring new areas of the domain to be investigated further. It is also notable that keywords such as climate change, $CO_2$ emissions and biomass have both high frequencies and citation bursts. It is evident that researchers across the globe have considered climate change and $CO_2$ emissions to be pivotal areas of concern in carbon emission research. Moreover, these co-words indicate the evolution of the carbon emission research domain. As indicated in Figure 9, the authors were more concerned with addressing the climate change-related aspects at the beginning. After identifying the contribution of greenhouse gases to climate change, the focus shifted towards greenhouse gases. As the researchers recognized fuel and behavioral aspects as key contributors to global GHG emissions, focus shifted towards perspectives related to fuel and behavior, which has resulted in a burst in "behavior" and "fuel". In the latter stages, the focus shifted towards production and technology related perspectives. It is also worth noting that researchers showed an interest in emissions trading

during the 2001–2008 period. The other notable aspect is the change of co-words over time due to the rapidly changing perspectives of carbon emission research.

| Keywords | Year | Strength | Begin | End | 1980 - 2019 |
|----------|------|----------|-------|-----|-------------|
| climate change | 1980 | 12.156 | 1999 | 2010 | |
| bioma | 1980 | 6.9543 | 2001 | 2014 | |
| emissions trading | 1980 | 3.9997 | 2001 | 2008 | |
| greenhouse gas | 1980 | 6.0641 | 2002 | 2013 | |
| cycle | 1980 | 5.5657 | 2005 | 2010 | |
| ecosystem | 1980 | 3.8947 | 2006 | 2009 | |
| storage | 1980 | 5.8279 | 2006 | 2012 | |
| behavior | 1980 | 3.8258 | 2008 | 2013 | |
| conservation | 1980 | 4.2931 | 2008 | 2012 | |
| fuel | 1980 | 5.2629 | 2008 | 2013 | |

**Figure 9.** Top 10 co-word citation bursts.

*4.4. Co-Citation Analysis Result*

Co-citation is another key measure of documents, which is defined as the frequency of two documents being cited together in other documents. Accordingly, a journal co-citation analysis, document co-citation analysis, and author co-citation analysis were conducted to identify the trends and patterns of carbon emission research. Co-citation analysis results are indicated in Figure 10 to understand the global trends of carbon emission research. Figure 10 describes the visual representation of networks of author co-citation analysis results. The top-ranked co-author by citation counts is Hertwich E.G. (2009), (cluster = #-1 and citation count = 68), followed by Peters G.P. (2011), (cluster #-1, citation count 60); Weidema B.P. (2008), (cluster #-1, citation count 59); Davis S.J. (2010), (cluster #-1, citation count 57); Pandey D. (2011), (cluster #-1, citation count 48) and Liu Z. (2015), (cluster #-1, citation count 42) on the sixth position. Meanwhile, Bond T.C. (2013) and Weber C.L. (2008), are ranked seventh with cluster #-1 and citation counts = 40, Finkbeiner M. (2009), (cluster #-1, citation count 39); Searchinger T. (2008), (cluster#-1, citation count 39) and Matthew H.S., (2008), (cluster #-1, citation count 38) are ranked eighth and tenth respectively.

The Intergovernmental Panel on Climate Change (IPCC) guidelines for national greenhouse gas inventories by Eggleston, et al. [88] was one of the most frequently cited documents, and according to google scholar citation reports, it has been cited approximately 1525 times. The study by Weidema, et al. [89], which investigated the ability to use carbon footprint as a catalyst for life cycle assessment was ranked second. The study of decomposition analysis on energy policy by Ang [90] also investigated the carbon emission of the energy sector. Lal [91] studied the carbon emission of farm operations. It is notable that carbon emission investigation of different industrial sectors remained to be of high interest amongst the authors and the above-mentioned studies have gained greater attention. It is also noted that evaluating carbon performance of different industrial sectors is an area of interest among researchers. In particular, carbon emission from various industries, such as energy and cement manufacturing were the center of focus among many authors across the globe.

Furthermore, Figure 11 revealed the top 10 core and active journals in carbon emission research domain to describe the network of co-citations based on authors citations and journal citations. Based on the frequency of citations of carbon emission related research, the top-ranked journal is "Energy Policy" with a citation count of (1313), the second is "Journal of Cleaner Production" (citation count = 1164) and the third is "Energy" (citation count = 849). Other journals in the same order from fourth to ten includes the "Environmental Science and Technology" (citation count = 828); "Ecological Economics" (citation count = 816); "Applied Economics" (citation count = 747); "Science" (citation count = 691);

"Renewable and Sustainable Energy Reviews" (citation count = 176); "International Journal of Life Cycle Assessment" (583), "Energy Economics" (citation count = 557) and "Nature" (citation count = 537).

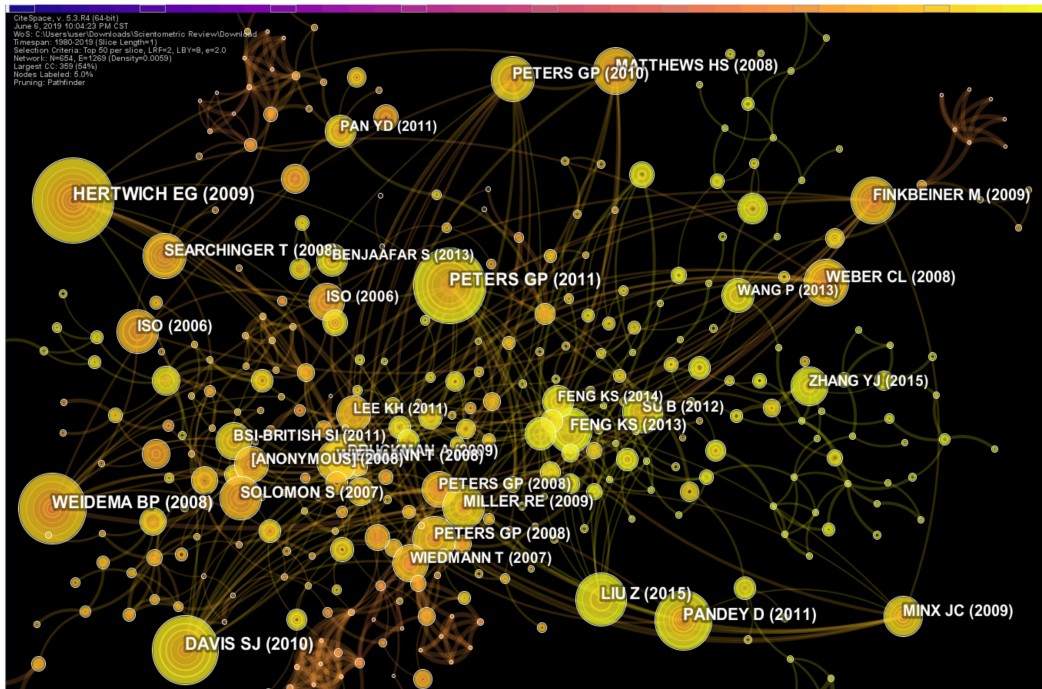

**Figure 10.** The network of co-citation analysis by author co-citation categories on carbon emission research.

### 4.5. Author Co-Citation Analysis

The relationships among the authors are revealed by an author co-citation analysis, which can recognize the evolution of the research communities. Accordingly, Figure 12 shows the author co-citation network, which comprises 220 nodes and 489 links. The size of the nodes represents the number of co-citations of each author and links between the nodes represents cooperative relations established amongst the authors. Accordingly, IPCC (frequency = 444, USA), Wiedmann T (frequency = 233, Australia), Peters GP (frequency = 185, Australia), Eggleston HS (frequency = 171, UK) and International Energy Agency (IEA) (frequency = 163) obtained the top five spots among the highly co-cited authors in carbon emission research. The diversity of author locations once again suggests that the attention on carbon emission research is gaining ground rapidly across the world.

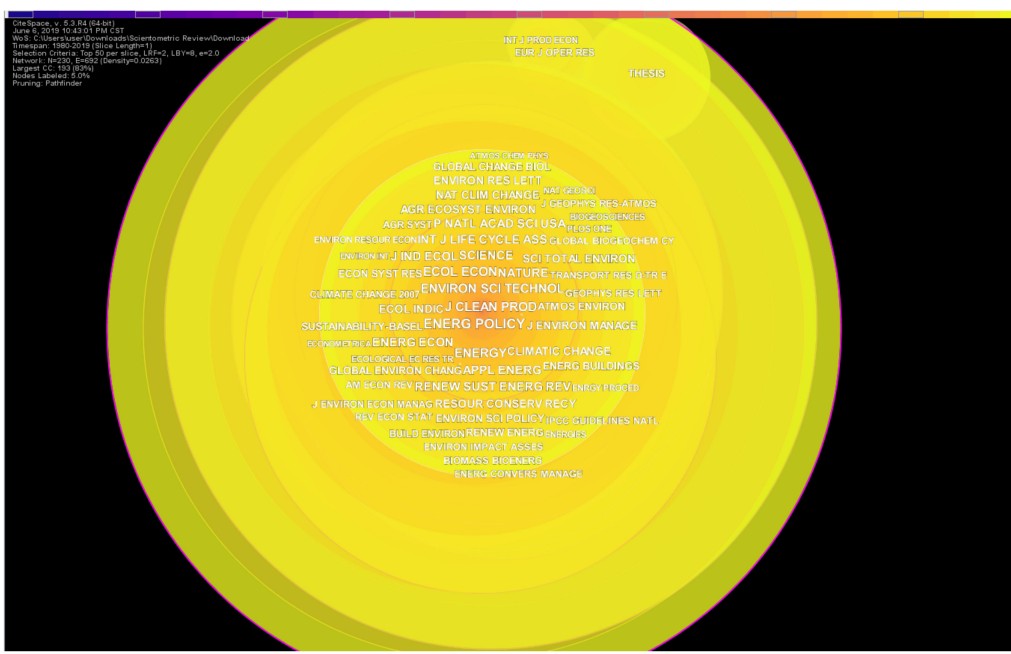

**Figure 11.** The network of co-citation analysis journal categories of carbon emission research.

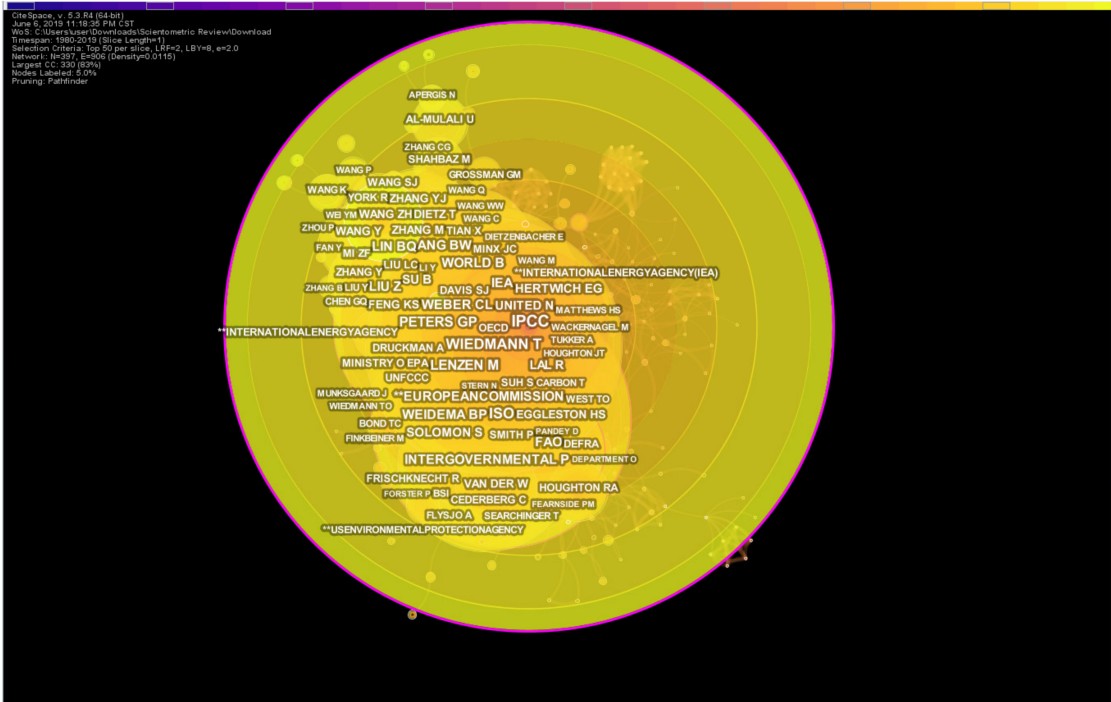

**Figure 12.** Author co-citation network.

The betweenness centrality of the authors is indicated by the purple rings of the nodes in Figure 12. Based on this metric, IPCC (centrality = 0.32), Peters GP (centrality = 0.27), Ang BW (centrality = 0.26), Wiedmann T (centrality = 0.23) and Houghton RA (centrality = 0.19) occupied the top five positions. These authors can be regarded as major sources of links in the domain. Even though highly cited authors might not receive a high betweenness centrality, an author with a high citation count and high betweenness centrality is likely to have a significant influence on carbon emission research. IPCC, Wiedmann T, and Peters GP have managed to obtain top spots in both citation count as well as the

betweenness centrality. Thus, these authors can be regarded as highly influenced researchers in the carbon emission research domain.

*4.6. Future Research Areas on Global Carbon Emissions*

Based on the findings of this review, scientometric analysis results, and discussion, several suggestions for future research work can be made. Scientometric analysis result identified a heavy focus on carbon emission related research in China. Despite China being the leading carbon emitter in the world, researchers should focus more on other countries such as the USA, Australia, and the UK. Moreover, the lack of research in developing countries was imminent. Therefore, more research studies on carbon emission related aspects should be conducted in developing countries. In particular, the researchers in China were predominantly focused on exploring the emission in the Chinese context. It is ideal for these researchers to expand their research studies to other countries to gain a better insight into global carbon emissions.

Life cycle estimation of carbon emissions has been the global trend of estimating carbon emissions. However, many researchers have indicated that life cycle estimation of carbon emissions neglect several uncertainties which may result in a significant variation of actual emissions and predicted emissions. Therefore, it is essential to explore newer approaches such as discrete event simulation and system dynamics. Moreover, the integration of information technology related tools such as Building Information Modelling and machine learning would provide more support to global researchers to estimate emissions more effectively.

The challenge of implementing carbon mitigation strategies is also obvious in the analysis, and it is identified as a gap between the research and practices. Several researchers have suggested carbon mitigation strategies but very few research studies have taken their practical application into consideration. Therefore, the practical implementation of these strategies needs further research focus. These research directions should be employed with the objectives of carbon estimation, identifying and implementing carbon mitigation strategies to improve the current situation.

## 5. Conclusions

Carbon emission research has received increasing global attention due to rapid global climate change. Thus, academics, international organizations, and government agencies have paid special consideration to identifying the carbon emission sources and thereby implementing various carbon mitigation strategies. The initial research on carbon emissions was conducted in 1981 and the domain has evolved significantly over the past two decades. This study provides a scientometric analysis of carbon emission research using 2561 bibliographic records extracted from the WoS core collection database. Several scientometric analysis, such as co-authorship, author co-citation, document co-citation, and journal co-citation analysis were utilized to identify and explore the trends in carbon emission research.

Research publication trend analysis revealed a steady increase in carbon emission research publications. Accordingly, the year 2018 was identified as the most productive year of publications, with a record of all-time high 443 publications. It is expected that the year 2019 will record an even higher number of publications as more and more researchers are now focusing on this important issue. As for the most productive authors on carbon emission research, several Chinese authors topped the list. While Yong Geng and Yi-Ming Wei were recognized as the most prolific authors in the domain, H. Scott Mathews was recognized as the only non-Chinese author among the top ten authors in the domain. The Peoples Republic of China, the USA, and England emerged as the most productive countries on carbon emission research. In terms of betweenness centrality, the USA topped the list, which indicated that the USA have had better research collaborations and links compared to other countries. On the contrary, betweenness centrality of China was low despite being the top publishing country. In addition, several Chinese institutions topped the list of the most productive institutions, which was led by the Chinese Academy of Sciences, Tsinghua University and the University of Chinese

Academy of Sciences. In terms of centrality, the University of Cambridge, University of Leeds and University of Colorado held the top positions. It is evident that despite the high number of publications, China is not recording good research connections with other countries on this domain. It was clear that the majority of the carbon emission research studies were conducted in developed countries. As a result of this, scientometric analysis returned most of the results related to carbon emissions based on the developed countries. However, the lack of studies on developing countries has resulted in bringing more attention to global researchers towards addressing this issue in developing countries.

Several core journals have published the most significant research studies on carbon emissions such as Energy Policy and Journal of Cleaner Production. These journals also received high citation frequencies as well as citation bursts making them the most prolific journals on carbon emission research. In terms of document co-citation analysis results, the publications by Eggleston HS and Weidema BP received the highest number of co-citations. Moreover, Solomon S and Finkbeiner M recorded significant citation bursts on carbon emission research. Most significantly, several citation bursts were recorded between 2010 and 2014, which indicate that more and more authors are focusing on carbon emission research since 2010. Additionally, IPCC and Wiedmann T topped the list of most cited authors in the domain.

According to the results observed through the scientometric analysis, it is evident that the carbon emission research domain has attracted the attention of global researchers. A significant increase in research publications over the past two years is a strong indication of the growth in the carbon emission research domain. Carbon capturing, predicting future carbon emissions through trend analysis, evaluating carbon performance, identifying carbon mitigation opportunities and ultimately achieving zero carbon emission goals are some of the most popular research areas in the carbon emission research domain. The scientometric analysis revealed the trends of carbon emission research over the past three decades which was the objective of this research.

There are certain limitations of this review which should be taken into consideration by the readers. Firstly, this paper was based only on the literature data obtained from the WoS core collection which might not cover all the available literature on the domain. Although many authors have used the WoS database, the comprehensive nature of this database cannot be assured. Moreover, data were only obtained from the journal articles which might not capture all the literature available on carbon emissions. It should also be noted that since the data obtained from the WoS core collection were filtered using the title, studies which might not reflect the carbon emission related work in the title were not included. However, this scientometric analysis provides a reflection of the global carbon emission research for researchers, government institutions and practitioners. It offers an in-depth understanding and a valuable insight into the most significant authors, institutions, countries in the carbon emission research domain as well as the trends of publications. The findings of this study can be used to obtain the necessary support and guidance to formulate carbon emission control policies.

**Author Contributions:** L.H.U.W.A. - Concept, analysis, discussion and write up; W.M.J. – Supervision; S.T.I. - data collection, methodology and writeup.

**Funding:** No funding.

**Conflicts of Interest:** No conflict of interest to be declared.

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
