# Peer review of "Global Research on Carbon Emissions: A Scientometric Review"

_sustainability, doi:10.3390/su11143972_

Reviewer 1 Report

The paper presents a comprehensive review of existing research on carbon emissions. The authors adopt some of the existing scientometric methods and tools used to assess performance of authors and spreading of topics of the research field. The work done is good but readability could be improved by shortening the “Results and Discussion” Section and by moving to an annex some of the details presented there such as those related to the most prolific authors. 

Another issue is about the several references in the text that are not included in the reference list at the end of the paper. I guess there are more than one hundred references but only 51 of them are included in the last section. See, for instance: 

- (Vernon, Wyman, Hubert, & Snyder, 1981) in page 1

- (Čuček, Klemeš, & Kravanja, 2012) in page 2  

- (Peters, 2010)

- Wiedmann and Minx (2008) in page 2

- Allen et al. (2009) in page 2

- West and Marland (2002) in page 2

- (Dietz, Gardner, Gilligan, Stern, & Vandenbergh, 2009) in page 3

- (Vernon et al., 1981) in page 7 

. and many other …

Author should check all the references. 

Other comments:

“Introduction” should include research gap, objective, research novelty, and paper outline. Currently these are in the “Literature review” section. 

In page 6:

“There are several science mapping tools are available to analyze insightful patterns and trends 237 of a scientific domain.” —>  “There are several science mapping tools available to analyze insightful patterns and trends 237 of a scientific domain.”

In page 8:

“For example, the lager nodes reflect ...” —> “For example, the larger nodes reflect …”

In page 10

- Authors should add the month of 2019 in the caption of Figure 3

- Authors should explain how “network pruning” is performed. 

- “heaily” —> “heavily”

In page 12

- Both Liu Y. And Geng Y. are the second most prolific authors. Is it correct? 

- “authr” —> “author”

- “cpncept” —> “concept”

In page 14.

Please define “citation bursts”.

Figure 12 is missing.

Author Response

Dear Reviewer,

Please find the attached document.

Thank you.

Reviewer 2 Report

The paper entitled "Global research on carbon emission: a scientometric review" aims at exploring the global publishing activity during the past decades on the greenhouse gases emission.

The manuscript presents serious flaws, starting from the bibliographic references, which are confused in the text, as well as missing in the bibliographic section (see for example Michaelowa and Michaelowa???; Agency 2017???; and so on). 

The main concern is: how could the Authors base their global research on a literature review and forget to correctly cite the references?

Additionally, the findings of the study led to foregone conclusions, which are of scarce interest for the readers.

Finally, I strongly suggest reading the instructions before submitting a manuscript, in order to follow the journal rules, starting from the Authors names and affiliations (Google Scholar?????).

Author Response

(The authors gave the same response as above.)

Reviewer 3 Report

ABSTRACT

It will be good if authors can point out other greenhouse gases (GHGs) such as nitrous oxide and methane before narrowing down to carbon – a highlight and an overview of other GHGs in abstract and introduction respectively will be good. This will provide more insight to readers that are not in this field. 

INTRODUCTION

Consider rewriting introduction in a clear, concise, and easy to read way with logical connection of the contents of one paragraph to another. Authors may want to connect paragraphs as:

Paragraph 1 and 2 - GHG emissions with examples including its sources and negative effects particularly adverse effects of climate change.

Paragraph 3 and 4 – Global carbon emissions and review studies carried out by scholars. This includes levels of emissions by various regions of the world, scholarly reviewed articles on global scale and the gap the present study will fill. Salient points in the literature review section can be summarised and brought into paragraph 3 and 4.

METHODOLOGY

The structure and contents looks good but will need improvement in spelling and grammar.

RESULTS AND DISCUSSIONS

Good presentation of results and discussions.

CONCLUSIONS

Conclusions looks more of a summary. Authors can conclude by outlining the major findings from the aim/objectives stated.

REFERENCES

Authors should adhere to journal’s referencing style. This includes all citations in the body of the work.

Author Response

Dear Reviewer,

Please find the attached document.

Thank you. 

Round  2

Reviewer 2 Report

The Authors have not addressed all the issues raised.

Thus, the study still presents flaws.

Author Response

Dear Reviewer,

 I would like to take this opportunity to thank you for taking your valuable time to review this paper.

I have modified the research paper based on your comments and thank you very much for your valuable comments to further develop this paper. Please find the actions taken for particular comments included below.

1. The manuscript presents serious flaws, starting from the bibliographic references, which are confused in the text, as well as missing in the bibliographic section (see for example Michaelowa and Michaelowa???; Agency 2017???; and so on).- References have been updated using the endnote software. Followed the guidelines of the journal when preparing the references. Thank you very much.

2. Structure and contentSome contents of the paper have been modified and updated. Conclusions and introduction have also been modified. Thank you very much. 

Once again thank you for your time and hoping to hear from you again.

Best regards.

Reviewer 3 Report

There      has been significant improvement following the first review of this      manuscript. However, more improvement is still required. Again, authors should consider rewriting      introduction in a clear, concise, and easy to read way with logical      connection of the contents of one paragraph to another. I have tried as      much as possible to suggest rephrased contents for the introduction in    this revised manuscript.

Authors      are yet to adhere to referencing style (format) of MDPI journals. For      ease, I have also provided the guide of the recommended referencing style      for this journal below:

In the text, reference numbers should be placed in square brackets [ ], and placed before the punctuation; for example [1], [1–3] or [1,3]. For embedded citations in the text with pagination, use both parentheses and brackets to indicate the reference number and page numbers; for example [5] (p. 10). or [6] (pp. 101–105).

References should be described as follows, depending on the type of work:

  Journal Articles:
1. Author 1, A.B.; Author 2, C.D. Title of the article. Abbreviated Journal Name YearVolume, page range. Available online: URL (accessed on Day Month Year).

  Books and Book Chapters:
2. Author 1, A.; Author 2, B. Book Title, 3rd ed.; Publisher: Publisher Location, Country, Year; pp. 154–196.
3. Author 1, A.; Author 2, B. Title of the chapter. In Book Title, 2nd ed.; Editor 1, A., Editor 2, B., Eds.; Publisher: Publisher Location, Country, Year; Volume 3, pp. 154–196.

  Unpublished work, submitted work, personal communication:
4. Author 1, A.B.; Author 2, C. Title of Unpublished Work. status (unpublished; manuscript in preparation).
5. Author 1, A.B.; Author 2, C. Title of Unpublished Work. Abbreviated Journal Name stage of publication (under review; accepted; in press).
6. Author 1, A.B. (University, City, State, Country); Author 2, C. (Institute, City, State, Country). Personal communication, Year.

  Conference Proceedings:
7. Author 1, A.B.; Author 2, C.D.; Author 3, E.F. Title of Presentation. In Title of the Collected Work (if available), Proceedings of the Name of the Conference, Location of Conference, Country, Date of Conference; Editor 1, Editor 2, Eds. (if available); Publisher: City, Country, Year (if available); Abstract Number (optional), Pagination (optional).

  Thesis:
8. Author 1, A.B. Title of Thesis. Level of Thesis, Degree-Granting University, Location of University, Date of Completion.

  Websites:
9. Title of Site. Available online: URL (accessed on Day Month Year).
Unlike published works, websites may change over time or disappear, so we encourage you create an archive of the cited website using a service such as WebCite. Archived websites should be cited using the link provided as follows:
10. Title of Site. URL (archived on Day Month Year).
Additional information can be found from this link (https://www.mdpi.com/journal/sustainability/instructions).

Author Response

Dear Reviewer,

 I would like to take this opportunity to thank you for taking your valuable time to review this paper.

I have modified the research paper based on your comments and thank you very much for your valuable comments to further develop this paper. Please find the actions taken for particular comments indicated below.

1. Authors should consider rewriting introduction in a clear, concise, and easy to read way with logical  connection of the contents of one paragraph to another- Introduction has been changed according to your suggestions. Thank you very much.

2. ReferencesReferences have been changed according to the guidelines. Thank you very much

Once again thank you for your time and hoping to hear from you again.

Best regards.

Round  3

Reviewer 2 Report

Even though the conclusions of the study are not original and, in my opinion, the contribution to the field is limited, the manuscript has significantly improved its quality, owing to the Reviewers' suggestions.

For these reasons, the manuscript is now suitable for the publication.